# Association of Hyperuricemia with 10-Year Atherosclerotic Cardiovascular Disease Risk among Chinese Adults and Elders

**DOI:** 10.3390/ijerph19116713

**Published:** 2022-05-31

**Authors:** Feilong Chen, Li Yuan, Tao Xu, Junting Liu, Shaomei Han

**Affiliations:** 1Department of Epidemiology and Statistics, Institute of Basic Medical Sciences, Chinese Academy of Medical Sciences & School of Basic Medicine, Peking Union Medical College, Beijing 100005, China; feilong_chen@ibms.pumc.edu.cn (F.C.); yuanli5665@gmail.com (L.Y.); hanshaomei.sd@gmail.com (S.H.); 2Department of Epidemiology, Capital Institute of Pediatrics, Beijing 100020, China

**Keywords:** hyperuricemia, atherosclerotic cardiovascular disease, serum uric acid, China-PAR project

## Abstract

The purpose of this cross-sectional study is to use a representative sample of the Chinese population and the atherosclerotic cardiovascular disease (ASCVD) risk assessment tool developed specifically for the Chinese to explore the prevalence of hyperuricemia (HUA) and the relationship between hyperuricemia and 10-year ASCVD risk in Chinese adults. Data was collected from the Chinese Physiological Constant and Health Condition survey. In this study, 12,988 subjects aged between 35 and 74 were selected by two-stage, cluster and random sampling. The sex-specific 10-year ASCVD risk scores equations, which were conducted by China-PAR project and specifically designed for Chinese adults, were used to assess the risk of developing ASCVD 10 years later. The ordinal regression model was used to explore the relationship between hyperuricemia and ASCVD risk. The total prevalence of hyperuricemia was 12.69%, and males’ prevalence was significantly higher than females (17.7% vs. 8.5%). Compared with people without hyperuricemia, the 10-year ASCVD risk scores of female with hyperuricemia increased significantly, but no significant increased happened in male. The ordinal regression model indicated that hyperuricemia subjects were 1.3 (males, 95% CI: 1.11–1.52) and 4.34 (females, 95% CI: 3.16–5.91) times more likely to increase their ASCVD risk levels than those without hyperuricemia. In conclusion, Hyperuricemia is prevalent among Chinese adults. In both genders, hyperuricemia was related with higher risk of 10-year ASCVD, and the relationship is much stronger in females than in males. Thus, sex-specific serum uric acid management and intervention strategies should be done in the prevention and control of cardiovascular events.

## 1. Introduction

Uric acid is the end product of purine metabolism in human liver and intestinal tissues, mainly excreted through the urinary tract [1,2]. Uric acid has the function of antioxidation and it is the vital antioxidants in human body. It participates in a variety of redox reactions and mediates type 2 immune responses [3,4]. However, the excessive increase of serum uric acid (SUA) will also result in potential pathological consequences. Hyperuricemia (HUA) is generally caused by an increase in the concentration of SUA, mostly because of the overproduction or underexcretion of uric acid, which leads to the accumulation of uric acid in the body [5]. With the rapid development of economy and society, the improvement of people’s lives and the increasing intake of high-protein and high-purine foods have both led to the increasing prevalence of hyperuricemia [6]. On the other hand, due to the sedentary lifestyle and the lack of physical exercise, the prevalence of chronic diseases such as obesity and type 2 diabetes has increased significantly and globally in recent years, further increasing the risk of metabolic syndrome. One of the outcomes is the increase in uric acid levels [7]. According to the American National Health and Nutrition Examination Survey (NHANES), the prevalence of hyperuricemia increased from 19.0% in 1988 to 21.5% in 2007 [8]. In China, a retrospective meta-analysis indicated that the prevalence of hyperuricemia in Chinese mainland was 22.7% for men and 11.0% for women in 2020, about 10% higher than a decade ago [9].

Cardiovascular diseases (CVDs) have become the leading causes of death both in China and all over the world, leading to serious disease burdens globally [10]. Hypertension, obesity, smoking etc. are all known risk factors of CVDs [11,12], but relationships between hyperuricemia and CVDs still remain controversial. A variety of studies have suggested that elevated SUA is an important risk factor for CVDs and also might be independent risk factors of CVDs and other kidney diseases [13,14,15,16]. Some epidemiological studies have pointed out that the hyperuricemia might be related to hypertension [17], metabolic syndrome [18], coronary artery disease [19] and cerebrovascular diseases [20]. Furthermore, there were also researches suggesting that SUA level could predict the development of hypertension, diabetes, obesity and kidney disease [21]. Meanwhile, some preliminary clinical trials have reported that lowering SUA level would be beneficial to the cardiovascular system [22]. However, there are also a considerable number of studies that hold opposing views. Some experts, such as the Framingham Heart Study groups, believe that uric acid is not a risk factor for CVDs and that clinicians should only rely on classic risk factors in patient assessment [23]. Nor have SUA been considered a cardiovascular risk factor by major professional societies [24,25].

Currently, some studies have used cardiovascular risk scores to explore the relationship between hyperuricemia and CVDs risks in Chinese adults and elders. However, most of these studies are based on population samples from some specific regions or cities, and there are few representative samples covering various regions and ethnic groups in China [26,27,28,29]. Secondly, most of the cardiovascular risk scores used in these studies are on the basis of foreign population cohorts, while lacking studies to verify the applicability of these risk scores to Chinese populations [30,31]. Our study selected the 10-year ASCVD risk scores constructed by the China-PAR project, which conducted representative sample cohorts based on Chinese people to build sex-specifics equations, and followed up for more than 10 years to verify the predictive ability of the equations, proved that they could well-predicted the CVD risk of Chinese people [32]. Therefore, the objective of this study was to analyze the association of hyperuricemia with 10-year ASCVD risk.

## 2. Materials and Methods

### 2.1. Study Population and Participants

The data was collected from the Chinese Physiological Constant and Health Condition (CPCHC) survey, a nationwide cross-sectional study based on the general Chinese population, which was conducted from 2007 to 2011. The study randomly selected six provinces or autonomous regions including Heilongjiang Province, Hunan Province, Yunnan Province, Inner Mongolia Autonomous region, Sichuan Province and Ningxia Hui Autonomous region, according to their geographical regionals, presence of ethnic minorities and economic characteristics. Then, the method of two-stage, random and cluster sampling was adopted. Firstly, 2–3 cities in each province are selected following simple random sampling on the basis of their economic status and the distribution of ethnic minorities. Secondly, the cluster random sampling method is used to select several communities in each city based on its population density. All people living in the selected communities were considered eligible for the survey if they were aged between 35–74 years, were not suffering from serious chronic diseases, had not run a high fever in the past 15 days, and agreed to sign the informed consent forms. Meanwhile, subjects who did not participate in laboratory tests and lack of serum uric acid concentration data, which was the main research index, would be excluded. The study population finally included 12,988 subjects who completed biochemical tests and body composition tests. After signing the informed consent forms, all the subjects volunteered to come to the temporary medical examination centers to participate in the survey. Before data collection, informed consent forms were obtained from each subject.

### 2.2. Data Collection and Interview

Data was collected by strictly trained investigators following standard procedures. Each subject was asked to fill out a standardized questionnaire in the form of a face-to-face interview. We gave each subjects prior training in the form of issuing leaflets and carrying out health education to make sure that they accurately understood the contents of the questionnaire. Physical examination items included height, weight, blood pressure, waist circumference (WC). Before measuring the blood pressure (BP), the subjects were required to rest for at least 10 min, and then to take the sitting position when measuring. The BP was measured three times using an Omron HEM-7000 electronic sphygmomanometer (Omron Healthcare; Muko, Kyoto, Japan) and the average values were taken as the final results. WC was measured with flexible and inelastic tape at the end of a gentle expiration. All clinical laboratory tests were conducted in the morning. The blood samples were drawn from subjects’ antecubital veins after at least 8-h, but no more than 12-h overnight fast. Total cholesterol (TC), high-density lipoprotein cholesterol (HDL-C) and uric acid were measured by Beckman AU series automatic biochemical analyzer (South Kraemer Blvd., Brea, CA, USA) and Sekiui Medical (Tokyo, Japan) reagents. Measurement of other blood biochemical tests, which were listed in the operation manual, were also used same instrument and Beckman AU reagents.

### 2.3. Definition of Covariates and Diagnostic Criteria

Covariates included demographic characteristics (sex, ethnicity, occupation, marital status and education level) and lifestyle factors such as drinking, exercise, cities, and subhealth status. Occupation was divided into mental work and physical work, the former was mainly based on the nervous system activities of the brain, and the latter depended on the movement of skeletal muscle system. Marital status was dichotomized into three groups: married, single and divorced or widowed. According to the educational background of the subjects, the education level was divided into three levels: illiterate or primary school level, secondary school (junior high school or senior high school) level, university and above level. Ethnic groups included Han, Miao, Yi, Mongolian, Korean, Hui, Tibetan, Tujia, and others. Drinking was defined as the drinking habit of the respondents during the survey, and exercise was defined as whether there were regular physical exercise habits in life. There were 20 areas including cities and rural areas based on the sampling location. Additionally, Hyperuricemia was defined as having a SUA greater than 416mmol/L in male and 357mmol/L in female [33,34] according to current clinical diagnostic criteria. Diabetes was defined as a condition in which FBG > 7 mmol/L and/or the current use of antidiabetic drugs [35,36].

### 2.4. Assessment of 10-Year ASCVD Risk Based on China-PAR Project

We used the ASCVD risk scores conducted by the China-PAR project to assess the 10-Year ASCVD risk [32]. 10-Year ASCVD risk scores, defined as the risk of developing the first ASCVD event over a 10-year period among a population without ASCVD at baseline, were built on the basis of large-scales nationwide prospective cohorts, and were used to predict subjects’ risk of developing ASCVD 10 years later through a series of demographic and serological indicators. There were two derivation cohorts to build the equations and two validation cohorts to verify their predictability. The projects included more than 120,000 subjects and were followed up for more than 10 years, and could better predict the ASCVD risk of Chinese adults. The sex-specific equation used age, treated or untreated systolic blood pressure, TC, HDL-C, smoking status and diabetes mellitus as common parameters for both men and women. At the same time, four additional variables of WC, geographical region, urbanization and family history of ASCVD were added to the male equation, and two additional variables of WC and geographical region were added to the females’, which further improves the predictability of the model. In this model, individual effect and mean effect were calculated using the sex-specific parameters mentioned above. The model was as below:(1)$$\mathrm{Ten}-\mathrm{year}\;\mathrm{ASCVD}\;\mathrm{Risk}\;\mathrm{Score}=1-S_{10}{}^{e^{\left(Individual\;effect-Mean\;effect\right)}}$$

*S*_10_ is a constant in men and women, which was the baseline survival rate for ASCVD at 10 years. In men and women, *S*_10_ equals to 0.97 and 0.99, respectively.

The sex-specific China-PAR equations had excellent performance of ASCVD risk prediction with good internal consistency and external validation. The results also demonstrated that the Pooled Cohort Equations (PCEs) in the American College of Cardiology/American Heart Association guideline were not appropriate for the Chinese population [32]. Additionally, the 10-year ASCVD risk scores were classified into three categories: scores less than 5% is considered as low risk, medium risk 5–10%, and high risk ≥ 10%.

### 2.5. Quality Control

Firstly, all the staffs were trained strictly before the survey, and they were required to strictly follow the standard operating procedures. Secondly, all the subjects received pre-education to ensure that they would actively cooperate with the researchers. Finally, we used the same brand and model of body composition analyzer, biochemical analyzer and sphygmomanometers at each center. They would be corrected every day before using. The biochemical laboratories involved in blood sample testing all followed the standardized common internal quality control program of Peking Union Medical College Hospital. Other detailed quality control measures could be seen elsewhere [37,38,39].

### 2.6. Statistical Analysis

Continuous normal distribution data were provided in forms of mean ± standard deviations and Student t tests were used for comparison between groups. Non-normal distribution data were described by medians and quartiles. Categorical data were summarized by numbers and percentages, and were compared using the Chi-square test or Kruskal-Wallis rank test. Ordinal regression models were used to explore the relationship between three-classified ASCVD risk levels and hyperuricemia. Odds ratios (OR) and 95% confidence intervals (CI) were calculated to estimate their association. Model 1 doesn’t adjust any confounders. Model 2 adjusted city, job, drinking, education levels, exercise and marital status. Two-tailed *p <* 0.05 was considered statistically significant. All the data were analyzed using SAS 9.4 software (SAS Institute Int., Cary, NC, USA).

## 3. Results

### 3.1. Characteristics of Participants and Prevalence of Hyperuricemi

This study included 12,988 participants aged between 34 and 74. The average age is 51.9 years for males and 51.3 years for females. Among them, 5976 were males (45.9%) and 7012 were females (54.1%). According to Table 1, age, SUA, education level, occupation, urbanization and ethnicity were both significantly different between groups in males and females. On the contrary, in female populations, there are statistical differences in marriage status and physical exercise between groups, while there are no such differences among males. For the risk of ASCVD, the distribution of 10-year ASCVD risk scores and risk levels show that there are statistically significances between groups in female population, but no significant differences are found in males. Meanwhile, compared with people without hyperuricemia, the 10-year ASCVD risk scores and risk levels of female with hyperuricemia both increased (All *p* for trend < 0.001), but not in men (*p* = 0.245). Therefore, the association between hyperuricemia and ASCVD is quite different in males and females.

As shown in Table 1, 1648 (12.69%) of the 12,988 subjects enrolled were diagnosed with hyperuricemia. Among them, the prevalence of males was significantly higher than females (17.7% vs. 8.5%). In the age subgroup, the prevalence of hyperuricemia increased with ages in the female population. The prevalence of hyperuricemia in suburban or rural areas was higher than in urban areas, and the result remains the same in both male and female populations. For males, the prevalence of Hui is the lowest and Miao is the highest. For females, the prevalence of Han is higher than Miao.

### 3.2. Distribution of 10-Year ASCVD Risk Levels among Demographic Characteristics

According to Table 2, for male population, 69.2% were low risk, 22.1% were medium risk, and 8.8% were high risk, and they were 94.3%, 5.1% and 0.6% among female, respectively. Male’s median 10-year ASCVD risk scores for each risk level were 1.79 (Low risk), 6.86 (Middle risk) and 12.68 (High risk), while female’s were 0.84 (Low risk), 6.35 (Middle risk) and 12.6 (High risk), respectively. Meanwhile, significant differences were found among risk levels group in both male and female population. The results also suggested that education levels, marriage status, exercise and occupation all showed statistically differences among different 10-Year ASCVD risk levels (*p* < 0.001). However, drinking habits were significantly different among groups only in male population, while urbanization only significant in females.

### 3.3. Association of Hyperuricemia with 10-Year China-PAR ASCVD Risk

The median ASCVD risk scores were 2.89% in males and 0.93% in females. Males had significantly higher ASCVD risk scores than female populations (*p* < 0.001). Table 3 indicated the results of the generalized linear models. In ordinal regression models, results were statistically insignificant when conducting one factor analysis for male population. After adjusting the confounders, it came to that hyperuricemia patients were 1.3 (OR:1.30, 95% CI: 1.11–1.52) times more likely to increase their ASCVD risk level by one or more levels than those without hyperuricemia. Results were significant even without adjusting any confounders in females, and the relationship was much stronger than males. Hyperuricemia females were 4.34 (adjusted OR:4.34, 95% CI: 3.16–5.91) times more likely to increase their ASCVD risk level by one or more levels. In summary, the risk of 10-year ASCVD in HUA group increased compared with non-HUA group.

## 4. Discussion

In this study, we used cross-sectional study method to explore the prevalence of hyperuricemia among Chinese adults and the relationship between hyperuricemia and the risk of 10-year ASCVD. We found that among Chinese adults, the prevalence of hyperuricemia varies greatly according to sex geographic region and nationality. At the same time, hyperuricemia was also associated with the increased 10-year risk of ASCVD, but this association varied significantly between two sexes. The effect of elevated uric acid on CVD in female population is greater than males.

Our study found that the concentration of SUA in Chinese adult males was higher than females. The prevalence of hyperuricemia was 17.7% for males and 8.5% for females, with a total prevalence of 12.69%. These results are different from the prevalence reported in a previous study. In that nationwide randomized, multicenter, multi-stage cross-sectional study conducted between 2009 and 2010, Liu et al. found that the prevalence of hyperuricemia in Chinese adults over the age of 18 was 9.9% in males and 7.0% in females, with a total prevalence of 8.4% [40]. On the one hand, this may be related to the composition of the sample population. This study included ethnic minorities from varied provinces, including Hui, Tibetan, Miao and so on, which can better reflect the actual situation of the Chinese population, while most of the samples selected in the above study are Han Chinese. On the other hand, our study included adults over the age of 35, while Liu et al. studied people over the age of 18, which may also account for the difference. In another meta-analysis, Liu et al. summarized the results of 33 Chinese and English articles published between 2000 and 2014 and concluded that the prevalence of men and women were 19.4% (95% CI: 17.6–21.1%) and 7.9% (95% CI: 6.6–9.3%), respectively [9]. The results are roughly consistent with our study. In terms of ethnic composition, the prevalence of hyperuricemia in Miao nationality is the highest and Hui is the lowest, which may due to their less consumption of meat and animal offal.

Hyperuricemia and CVD are both very common chronic diseases in the world, and their prevalence rates are on the rise year by year [41,42,43]. There have been many previous studies on the relationship between SUA and CVDs, but the final conclusion still remains controversial. Most epidemiological studies have shown that elevated SUA levels are associated with CVD, such as coronary heart disease, stroke, congestive heart failure, etc., and an increase risk for mortality due to CVD in general population and patients with confirmed CHD. Kannel et al. [43] followed 5127 participants for about 12 years in the context of the Framingham study in 1967. The results indicated that hyperuricemia was associated with an increased risk of coronary heart disease in males between the ages of 30 and 59. J. Fang et al. followed up 5926 adults aged 25–74 years from 1971 to 1992, suggesting that the increase of uric acid level was positively correlated with CHD-related mortality, and the results were consistent not only in men and women, but also in the black and the white [14]. In another prospective cohort study of Asians, 90,393 (>35 years of age) adults (41,879 men and 48,541 women) were followed up over a mean of 8.2 years. The results showed that hyperuricemia and adjusted all-cause mortality, total CVDor stroke risk increased significantly by 16%, 39% and 35%, respectively [44]. Our study is consistent with the above studies, after controlling the urban clustering effect and correcting confounding factors, hyperuricemia in both sexes will lead to an increase in 10-year ASCVD risk. Compared with non-HUA group, HUA group increased the risk by 30% (OR: 1.30, 95% CI: 1.11, 1.52) for males and 334% (OR: 4.34, 95% CI: 3.16, 5.91) for females. However, there are also some studies failing to demonstrated a significant association between elevated hyperuricemia and cardiovascular events or mortality. In the famous Framingham Cardiovascular study [23], 6763 subjects with an average age of 47 years were followed up for 117,367 person-years. After adjusting for confounding factors including age and traditional cardiovascular risk factors, no relationship between elevated uric acid levels and CVDs and all-cause mortality was found in both male and female populations. In addition, in the Third National Health and Nutrition Examination Survey (NHANES III) study, the researchers followed 11,009 subjects for up to 14.5 years (median). The results also did not show an association between SUA levels and CVD or CHD-related mortality. Of note, adding UA to the predictive model with traditional cardiovascular risk factors did not lead to a significant increase in C statistics, receiver-operating characteristic (ROC)-area under curve or the net reclassification index (NRI), suggesting that UA did not provide prognostic information provided by traditional cardiovascular risk factors in predicting CHD-related deaths [45]. In summary, there is still no definitive explanation for the association between uric acid levels and cardiovascular events, which may be due to its frequent and intricate relationship with other cardiovascular risk factors, as well as conflicting results from epidemiological or clinical studies on the association of UA with CVD.

In this study, the association between uric acid levels and ASCVD also showed sex differences. The effect that hyperuricemia increased their ASCVD risk is much stronger in females than in males. The difference has also been reported in previous studies. In the NHANES I study (a cross-sectional population-based study of 5926 subjects 25–74 years of age followed up for a mean of 16.4 years), the results showed that the increase SUA was associated with CHD-related mortality in men and women, with a RR of 1.77 (95% CI: 1.08–3.980) in males and 3.00 (95% CI: 1.45–6.28) in females for 4th vs. 1st UA quartiles [14]. Another meta-analysis conducted by LI et al. (included 29 prospective cohort studies, including 958,419 subjects) found that each 1mg/dl increase in SUA, the risk of CHD mortality increased in women (RR = 2.44) but not in men without heterogeneity in analyses [46]. This may be related to the mechanisms of cardiovascular damage caused by uric acid. Experimental and clinical studies have confirmed that the mechanisms of harmful effects of elevated uric acid on cardiovascular health, including increased oxidative stress [4,47,48,49], decreased availability of nitric oxide and endothelial dysfunction [50], promotion of local and systemic inflammation, vasoconstriction and proliferation of vascular smooth muscle cell, insulin resistance [51,52] and metabolic dysregulation [18,53]. In women, as the secretion of estrogen decreases with age, its protective effect on the cardiovascular system may gradually decrease, so it is more likely to be affected by elevated uric acid levels.

This study has certain strengths. Firstly, we used a representative sample of the whole Chinese population. Secondly, researchers formulated strict quality control measures to ensure the authenticity. In addition, this study used the 10-year ASCVD risk scores constructed by the China-PAR project as a tool to assess the risk of atherosclerotic cardiovascular diseases. This study is the first to use this score and a representative sample of the Chinese population to explore the relationship between hyperuricemia and ASCVD risk. However, this study also has some limitations. First, this is a cross-sectional study, and the sequence of exposure and outcome is difficult to determine, so it is restricted in the process of causal inference. What’s more, since many factors were reported by the subjects themselves, such as disease history, smoking and drinking status, family history, the reporting of the above problems may deviate from the actual situation, which may result in un-avoided memory bias.

## 5. Conclusions

In conclusion, hyperuricemia is prevalent among Chinese adults and the total prevalence is 12.69%. It appears that higher serum uric acid levels are associated with the 10-year ASCVD risk. In both sexes, hyperuricemia was related with higher risk of 10-year ASCVD, but the relationship is much stronger in females than in males, which suggests that females are more likely to be affected by hyperuricemia. Thus, sex-specific serum uric acid management and intervention strategies should be done in the clinical prevention and control of cardiovascular events.

## Figures and Tables

**Table 1 ijerph-19-06713-t001:** Baseline information of participants and prevalence of hyperuricemia by sex.

Variables	Male		*p*	Female		*p*
	Non-HUA	HUA		Non-HUA	HUA	
	*n* = 4921	*n* = 1055		*n* = 6419	*n* = 593	
SUA(μmol/L)	318.6 (272.8, 359.1)	459.0 (434.0, 496.0)	<0.001	247.4 (209.7, 286.0)	388.8 (369.3, 421.1)	<0.001
10-Year ASCVD risk score (%)	2.83 (1.25, 5.88)	3.19 (1.30, 6.27)	0.245	0.87 (0.35, 1.93)	1.89 (0.82, 3.58)	<0.001
10-Year ASCVD risk group			0.142			<0.001
Low risk	3431 (83.0)	703 (17.0)		6113 (92.4)	500 (7.6)	
Middle risk	1068 (81.0)	251 (19.0)		276 (77.5)	80 (22.5)	
High risk	422 (80.7)	101 (19.3)		30 (69.8)	13 (30.2)	
Education			<0.001			0.004
Primary School and blow	1155 (84.3)	215 (15.7)		2039 (90.4)	217 (9.6)	
Middle School	2215 (82.9)	457 (17.1)		2652 (91.4)	248 (8.6)	
University and above	1365 (79.1)	360 (20.9)		1552 (93.4)	110 (6.6)	
Unknown	186 (89.0)	23 (11.0)		176 (90.7)	18 (9.3)	
Marriage			0.313			<0.001
Married	4492 (82.0)	987 (18.0)		5623 (92.2)	479 (7.8)	
Single	65 (87.8)	9 (12.2)		54 (93.1)	4 (6.9)	
Divorced or widowed	131 (84.5)	24 (15.5)		490 (87.2)	72 (12.8)	
Unknown	233 (86.9)	35 (13.1)		252 (86.9)	38 (13.1)	
Drink			<0.001			0.374
Yes	2315 (78.4)	638 (21.6)		355 (90.3)	38 (9.7)	
No	2606 (86.2)	417 (13.8)		6064 (91.6)	555 (8.4)	
Occupation			<0.001			0.010
Mental job	1987 (80.3)	489 (19.7)		2109 (92.8)	164 (7.2)	
Physical job	2934 (83.8)	566 (16.2)		4310 (90.9)	429 (9.1)	
Exercise			0.076			0.022
Yes	1426 (82.2)	309 (17.8)		1715 (91.5)	160 (8.5)	
No	2600 (84.2)	489 (15.8)		3582 (93.2)	263 (6.8)	
Unknown	895 (77.7)	257 (22.3)		1122 (86.8)	170 (13.2)	
Urbanization			0.016			<0.001
Urban areas	1797 (83.9)	344 (16.1)		2623 (93.0)	198 (7.0)	
Suburban and rural areas	3122 (81.5)	711 (18.5)		3784 (90.5)	395 (9.5)	
Age group			0.017			<0.001
35–44 years	1152 (80.8)	369 (19.2)		2269 (95.7)	103 (4.9)	
45–54 years	1414 (83.9)	271 (16.1)		1927 (92.8)	150 (7.2)	
55–64 years	1189 (83.8)	230 (16.2)		1517 (89.4)	180 (10.6)	
65–74 years	766 (80.5)	185 (19.5)		706 (81.5)	160 (18.5)	
Race			<0.001			<0.001
Han	3103 (81.4)	708 (18.6)		4037 (90.3)	436 (9.7)	
Yi	493 (84.4)	91 (15.6)		745 (93.2)	54 (6.8)	
Miao	91 (74.0)	32 (26.0)		127 (94.1)	8 (5.9)	
Mongolia	230 (81.6)	52 (18.4)		391 (94.9)	21 (5.1)	
Tibetan	123 (81.5)	28 (18.5)		132 (93.6)	9 (6.4)	
Korean	157 (79.3)	41 (20.7)		296 (91.9)	26 (8.1)	
Hui	503 (92.6)	40 (7.4)		446 (95.5)	21 (4.5)	
Tujia	171 (78.4)	47 (21.6)		170 (93.9)	11 (6.1)	
Other	50 (75.8)	16 (24.2)		75 (91.5)	7 (8.5)	
Province			<0.001			<0.001
Sichuan	659 (13.4)	228 (21.6)		903 (14.1)	152 (25.6)	
Heilongjiang	665 (13.5)	181 (17.2)		1127 (17.6)	91 (15.3)	
Hunan	819 (16.6)	143 (13.6)		826 (12.9)	54 (9.1)	
Inner Mongolia	932 (18.9)	222 (21.0)		1298 (20.2)	137 (23.1)	
Yunnan	941 (19.1)	203 (19.2)		1343 (20.9)	123 (20.7)	
Ningxia	905 (18.4)	78 (7.4)		922 (14.4)	36 (6.1)	

Abbreviation: SUA, serum uric acid; ASCVD, atherosclerotic cardiovascular disease; HUA, hyperuricemia.

**Table 2 ijerph-19-06713-t002:** Distribution of ASCVD risk levels among demographic characteristics by sex.

Variables	Male				Female			
	Low Risk	Middle Risk	High Risk	*p*	Low Risk	Middle Risk	High Risk	*p*
Total	*n* = 4134 (69.2)	*n* = 1319 (22.1)	*n* = 523 (8.8)		*n* = 6613 (94.3)	*n* = 356 (5.1)	*n* = 43 (0.6)	
10-Year ASCVD risk score (%)	1.79 (0.90, 3.07)	6.86 (5.88, 8.21)	12.68 (11.20, 15.46)	<0.001	0.84 (0.35, 1.81)	6.35 (5.58, 7.35)	12.6 (10.85, 14.00)	<0.001
Education				<0.001				<0.001
Primary School & blow	907 (66.2)	303 (22.1)	160 (11.7)		2090 (92.6)	147 (6.5)	19 (0.8)	
Middle School	1792 (67.1)	635 (23.8)	245 (9.2)		2710 (93.4)	171 (5.9)	19 (0.7)	
University & above	1311 (76.0)	319 (18.5)	95 (5.5)		1633 (98.3)	24 (1.4)	5 (0.3)	
Unknown	124 (59.3)	62 (29.7)	23 (11.0)		180 (92.8)	14 (7.2)	0 (0.0)	
Marriage				<0.001				<0.001
Married	3891 (71.0)	1153 (21.0)	435 (7.9)		5854 (95.9)	221 (3.6)	27 (0.4)	
Single	55 (74.3)	10 (13.5)	9 (12.2)		52 (89.7)	5 (8.6)	1 (1.7)	
Divorced & widowed	81 (52.3)	54 (34.8)	20 (12.9)		467 (83.1)	84 (14.9)	11 (2.0)	
Unknown	107 (39.9)	102 (38.1)	59 (22.0)		240 (82.8)	46 (15.9)	4 (1.3)	
Drink				<0.001				0.620
No	2032 (67.2)	666 (22.0)	325 (10.8)		6238 (94.2)	340 (5.1)	41 (0.6)	
Yes	2102 (71.2)	653 (22.1)	198 (6.7)		375 (95.4)	16 (4.1)	2 (0.5)	
Occupation				<0.001				<0.001
Physical job	2185 (62.4)	914 (26.1)	401 (11.5)		4396 (92.8)	307 (6.5)	36 (0.8)	
Mental job	1949 (78.7)	405 (16.4)	122 (4.9)		2217 (97.5)	49 (2.2)	7 (0.3)	
Exercise				<0.001				<0.001
No	2390 (77.4)	508 (16.4)	191 (6.2)		3691 (96.0)	136 (3.5)	18 (0.5)	
Yes	1045 (60.2)	461 (26.6)	229 (13.2)		1682 (89.7)	171 (9.1)	22 (1.2)	
Unknown	699 (60.7)	350 (30.4)	103 (8.9)		1240 (96.0)	49 (3.8)	3 (0.2)	
Urbanization				<0.001				0.465
Suburban & rural areas	2606 (68.0)	843 (22.0)	384 (10.0)		3942 (94.3)	216 (5.2)	21 (0.5)	
Urban areas	1527 (71.3)	475 (22.2)	139 (6.5)		2661 (94.3)	139 (4.9)	21 (0.7)	

**Table 3 ijerph-19-06713-t003:** Association of hyperuricemia with 10-Year ASCVD risk levels by sex.

Model	Male		Female	
	Non-HUA	HUA	Non-HUA	HUA
The Ordinal Regression Models (OR and 95% CI)				
Model 1	1 (reference)	1.14 (0.99, 1.31)	1 (reference)	3.85 (2.97, 4.94) *
Model 2	1 (reference)	1.30 (1.11, 1.52) *	1 (reference)	4.34 (3.16, 5.91) *

Notes: Model 1 not adjust any confounders; Model 2 further adjusted city, job, urbanization, marriage, education, drinking, exercise. HUA: hyperuricemia. * *p* < 0.05.

## Data Availability

The data underlying this article will be shared on reasonable request to the corresponding author.

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
