# Peer review of "Association of Hyperuricemia with 10-Year Atherosclerotic Cardiovascular Disease Risk among Chinese Adults and Elders"

_ijerph, 2022, doi:10.3390/ijerph19116713_

Round 1

Reviewer 1 Report

-          From the title of the study, it looks like the study is a prospective and observational cohort investigation to evaluate the association between the risk factors and hyperuricemia for ten years. However, the study uses a tool called “the 10-year ASCVD risk scores”. Is that correct? This needs to be clarified for the readers. Please explain the validity and reliability of the tool.

-          Lines 38-39 are vague. Please be more specific on how the development of the economy is related to the prevalence of hyperuricemia.

-          Line 85-86: Please clarify the following statement. What were the requirements /inclusion criteria?

“People over 35 years old who met the 85 requirements in each community were considered eligible as study subjects”

-          The results of this study could have been influenced by confounding variables such as triglycerides, smoking, waist circumference, blood pressure, etc. The readers can raise the question of why the analysis was not adjusted for the above variables.

-          Lines:287-88. Wording consideration is required. Was the focus of this study on sexual differences or sex differences?

Author Response

Dear reviewer,

Many thanks for your comments and advices. These are my response to the comments.

Reviewer #1: 

  1. From the title of the study, it looks like the study is a prospective and observational cohort investigation to evaluate the association between the risk factors and hyperuricemia for ten years. However, the study uses a tool called “the 10-year ASCVD risk scores”. Is that correct? This needs to be clarified for the readers. Please explain the validity and reliability of the tool.

Response1:Thanks for your careful comments.

I'm sorry to have caused you confusion. Our study is a cross-sectional study, which aims to explore the association between hyperuricemia and 10-year ASCVD risk among Chinese adults and elders. The 10-year ASCVD risk scores, which were a predictive model, were based on large-scale nationwide prospective cohorts and they were used to predict subjects' risk of developing ASCVD 10 years later through a series of demographic and serological indicators. There were two derivation cohorts to build the equations and two validation cohorts to verify their predictability. The projects included more than 120,000 subjects and were followed up for more than 10 years. The sex-specific China-PAR equations had excellent performance of ASCVD risk prediction with good internal consistency and external validation. The results also demonstrated that the Pooled Cohort Equations (PCEs) in the American College of Cardiology/American Heart Association guideline were not appropriate for the Chinese population [32]. We have added this explanation to the manuscript in Line 137-150.

  1. Yang X, Li J, Hu D, Chen J, Li Y, Huang J, Liu X, Liu F, Cao J, Shen C et al: Predicting the 10-Year Risks of Atherosclerotic Cardiovascular Disease in Chinese Population: The China-PAR Project (Prediction for ASCVD Risk in China). Circulation 2016, 134(19):1430-1440.

  1. Lines 38-39 are vague. Please be more specific on how the development of the economy is related to the prevalence of hyperuricemia.

Response2:

Thanks for your advice. I have expanded the introduction on the association between economic development and the increasing prevalence of hyperuricemia in the manuscript in Line 35-40. The detailed text and references as follow:

“With the rapid development of economy and society, the improvement of people's lives and the increasing intake of high-protein and high-purine foods both lead to the increasing prevalence of hyperuricemia [6]. On the other hand, due to the sedentary lifestyle and the lack of physical exercise, the prevalence of chronic diseases such as obesity and type 2 diabetes has increased significantly and globally in recent years, further increasing the risk of metabolic syndrome [7]. One of the outcomes is the increase in uric acid levels.”

  1. Pascart T, Lioté F: Gout: state of the art after a decade of developments. Rheumatology (Oxford, England) 2019, 58(1):27-44.
  2. Roubenoff R, Klag MJ, Mead LA, Liang KY, Seidler AJ, Hochberg MC: Incidence and risk factors for gout in white men. Jama 1991, 266(21):3004-3007.

3.  Line 85-86: Please clarify the following statement. What were the requirements /inclusion criteria?

“People over 35 years old who met the 85 requirements in each community were considered eligible as study subjects”

Response3:

Thanks for your comment. I have added inclusion and exclusion criteria to the manuscript in Line 86-91. The detailed information as follow:

“All people living in the selected communities were considered eligible for the survey if they were aged between 35-74 years old, were not suffering from serious chronic diseases, had not run a high fever in the past 15 days, and agreed to sign the informed consent forms. Meanwhile, subjects who did not participate in laboratory tests and lack of serum uric acid concentration data, which were the main research index, would be excluded.”

  1. The results of this study could have been influenced by confounding variables such as triglycerides, smoking, waist circumference, blood pressure, etc. The readers can raise the question of why the analysis was not adjusted for the above variables.

Response4: Thanks for your comment.

Smoking, waist circumference, blood pressure, total cholesterol and high-density lipoprotein (HDL-C), were used in the construction of ASCVD risk scores, which were one of the main evaluation indicators of our study. In the meanwhile, it is known that triglycerides, total cholesterol and high-density lipoprotein are highly correlated [32]. Therefore, to avoid the possible collinearity problem, we did not adjust triglycerides, smoking, waist circumference, blood pressure and other blood lipid markers when exploring the association between serum uric acid and 10-year ASCVD risk scores.

  1. Yang X, Li J, Hu D, Chen J, Li Y, Huang J, Liu X, Liu F, Cao J, Shen C et al: Predicting the 10-Year Risks of Atherosclerotic Cardiovascular Disease in Chinese Population: The China-PAR Project (Prediction for ASCVD Risk in China). Circulation 2016, 134(19):1430-1440.

  1. Lines:287-88. Wording consideration is required. Was the focus of this study on sexual differences or sex differences?

Response5: Thanks for your reminding.

It is a spell mistake. I have corrected the expression as “sex differences”.

Thank you for your kind attention and looking forwards to your favorable reply.

Yours sincerely,

Feilong Chen

Tao Xu

Department of epidemiology and statistics

Institute of Basic Medical Sciences, Chinese Academy of Medical Sciences & School of Basic Medicine, Peking Union Medical College

5, Dong dan san tiao

Beijing 100005

China

Tel: 86 10 69156408

Reviewer 2 Report

Chen etc summarized the prevalence of hyperuricemia in China and the relationship between hyperuricemia and 10 year ASCVD risk using China-PAR. Overall, the article is well-written. Here are a few comments.

1.    Authors used negative binomial and ordinal regression. Reviewer understands they have shown with continuous risk score and categorical score. However, the result of negative binomial is only line 202-207, and it may be confusing to readers to have two models without any explorations. Reviewer suggests to omit the results of negative binomial in the manuscript, which probably does not change author’s claim

2.    Line 85 shows the age criteria is older than 35, which meets China-PAR. How about the maximum age? There are certain percentages of older than 65 in Table1. Does that indicate 65-74? Author should add explanation in the manuscript.

3.    What is HUA? Verify in either manuscript or abbreviation.

4.    Verify the difference of between the number of Table1(Male=6045, female=7127) and the number of Table2(Male=5976, female=7012).

5.    China-PAR use 4 risk classification(<5%, 5%-7.4%,7.5%-9.9% and ≧10%). Verify why authors use three categories.

Author Response

Dear reviewer,

Many thanks for your comments and advices. These are my responses to the comments.

Reviewer #2: 

  1. Authors used negative binomial and ordinal regression. Reviewer understands they have shown with continuous risk score and categorical score. However, the result of negative binomial is only line 202-207, and it may be confusing to readers to have two models without any explorations. Reviewer suggests to omit the results of negative binomial in the manuscript, which probably does not change author’s claim.

Response1: Thanks for your valuable advice.

We initially hoped to explore the association between uric acid and 10-year ASCVD risk in terms of continuous and classified risk scores. Indeed, we did not interpret the results of the negative binomial model too much, which might cause some confusion to the readers. We have deleted the content about the negative binomial model in the manuscript according to your suggestion.

  1. Line 85 shows the age criteria is older than 35, which meets China-PAR. How about the maximum age? There are certain percentages of older than 65 in Table1. Does that indicate 65-74? Author should add explanation in the manuscript.

Response2: Thanks for your advice.

Our study included the participants aged between 35 and 74. It was true that variable age group (≥65 years) might cause readers confusion. We have changed the expression into (65-74 years). Meanwhile, I described the range and mean values of subjects’ age by sex in Line 174.

  1. What is HUA? Verify in either manuscript or abbreviation.

Renponse3: Many thanks for your reminding.

HUA means “hyperuricemia”. Hyperuricemia (HUA) was defined as having a SUA greater than 416mmol/L in male and 357mmol/L in female according to current clinical diagnostic criteria. I have defined the abbreviation in the manuscript and table.

  1. Verify the difference of between the number of Table1 (Male=6045, female=7127) and the number of Table2 (Male=5976, female=7012).

Renponse4: Thanks for your comment.

Table1 showed the baseline information of all participants. Table2 indicated the distribution of ASCVD risk levels among demographic characteristics and the risk levels were evaluated through 10-year ASCVD risk scores. The sex-specific equation used age, treated or untreated systolic blood pressure, TC, HDL-C, smoking status and diabetes mellitus as common parameters for both men and women. Meanwhile, four additional variables of waist circumference (WC), geographical region, urbanization and family history of ASCVD were added to the male equation, and two additional variables of WC and geographical region were added to the females’, which further improves the predictability of the model. However, some subjects had missing data in the indicators needed to build the equation, so they could not get a 10-year ASCVD risk score. A total of 184 people (1.4% of total participants) failed to get the 10-year ASCVD risk scores, which should be considered incomplete data. Therefore, we deleted these incomplete data and rewrote the table 1 and related results description to assure the same number (Total number=12,988, Male=5976, female=7012) in table 1 and table 2.

  1. China-PAR use 4 risk classification(<5%, 5%-7.4%,7.5%-9.9% and â‰§10%). Verify why authors use three categories.

Response5: Thank you for your comment.

China-PAR project used 4 risk classifications on the basis of clinical meaningful cut-off points. They mainly focus on the clinical meaning [32]. Our project paid more attention to the association between serum uric acid levels and 10-year ASCVD risk to provide scientific evidence for early intervention in patients with high uric acid to reduce their risk of cardiovascular disease. From the perspective of public health and behavioral intervention, there is no need for over-detailed risk classification, otherwise it will complicate the implementation of intervention measures. Therefore, we combine 5%-7.4% group and 7.5%-9.9% group into one level as a medium-level risk.

  1. Yang X, Li J, Hu D, Chen J, Li Y, Huang J, Liu X, Liu F, Cao J, Shen C et al: Predicting the 10-Year Risks of Atherosclerotic Cardiovascular Disease in Chinese Population: The China-PAR Project (Prediction for ASCVD Risk in China). Circulation 2016, 134(19):1430-1440.

Thank you for your kind attention and looking forwards to your favorable reply.

Yours sincerely,

Feilong Chen

Tao Xu

Department of epidemiology and statistics

Institute of Basic Medical Sciences, Chinese Academy of Medical Sciences & School of Basic Medicine, Peking Union Medical College

5, Dong dan san tiao

Beijing 100005

China

Tel: 86 10 69156408

Round 2

Reviewer 1 Report

Dear Authors,

Thanks for addressing the comments provided for the previous version. Readers would appreciate it if elaboration is provided regarding how risk scores were calculated for the population of this study and determined whether the scores were categorized as less than 5% ( low risk), 5-10% (medium risk), and ≥ 10% (high risk). Perhaps providing a table would help.

The data provided in Tables 1 and 2, looks like descriptive data. If so, the total numbers for each sectional row should be equal to the other sectional rows for each column, but it is not the case for all of them. For example, in Table 1, the total number related to drinks (yes and no) is not equal to the total number associated with marriage (married, single, divorce)  for the Non-HUA column. Based on the information provided in the first-row corresponding to the sample size for this column, the total number should be n=4,921. Please clarify otherwise. Thank you

Author Response

Dear reviewer,

Many thanks for your comments and advices. These are my responses to the comments.

Reviewer #1: 

  1. Thanks for addressing the comments provided for the previous version. Readers would appreciate it if elaboration is provided regarding how risk scores were calculated for the population of this study and determined whether the scores were categorized as less than 5% (low risk), 5-10% (medium risk), and ≥ 10% (high risk). Perhaps providing a table would help.

Response: Many thanks for your comments. 10-year ASCVD risk was defined as the risk of developing the first ASCVD event over a 10-year period among a population without ASCVD at baseline [1]. The model of 10-year ASCVD risk score was similar to Framingham CVD scores, except that the former was specially designed for Chinese adults. I have shown the formula for calculating the sex-specific 10-year ASCVD risk scores in the revised manuscript and provided references. In this model, individual effect and mean effect were calculated using the sex-specific parameters mentioned above. The model was as below:

S10 is a constant in men and women, which was the baseline survival rate for ASCVD at 10 years. In men and women, S10 equals to 0.97 and 0.99, respectively. The detailed model parameters and calculation examples are shown in Supple table 1 and Supple table 2:

Supple Table 1. Gender-specific Parameters of Equations for Predicting 10-year ASCVD Risk with Specific Example

Male

Female

Coefficient

Individual Example Value

Coefficient×Value

Coefficient

Individual Example Value

Coefficient×Value

Example: 60 years of age with untreated 130mmHg, total cholesterol 210mg/dL, HDL-C 55 mg/dL, waist circumference 80cm, nonsmoking, diabetes, living in urban area of Northern China, and without family history of ASCVD.

Ln(age), y

31.97

4.09

130/88

24.87

4.09

101.84

Ln(treated SBP), mmHg

27.39

-

-

20.71

-

-

Ln(untreated SBP), mmHg

26.15

4.87

127.28

19.98

4.87

97.26

Ln(total cholesterol), mg/dL

0.62

5.35

3.32

0.06

5.35

0.31

Ln(HDL-C), mg/dL

-0.69

4.01

-2.78

-0.22

4.01

-0.87

Ln(waist circumference), cm

-0.71

4.38

-3.12

1.48

4.38

6.46

Current smoker (1=Yes, 0=No)

3.96

0

0.00

0.49

0

0.00

Diabetes (1=Yes, 0=No)

0.36

1

0.36

0.57

1

0.00

Geographic region (1=Northern China, 0=Southern China)

0.48

1

0.48

0.54

1

0.54

Urbanization (1=Urban, 0=Rural)

-0.16

1

-0.16

N/A

N/A

N/A

Family history of ASCVD (1=Yes, 0=No)

6.22

0

0.00

N/A

N/A

N/A

Ln(age)* Ln(treated SBP)

-6.02

-

-

-4.53

-

-

Ln(age)* Ln(untreated SBP)

-5.73

19.93

-114.21

-4.36

19.93

-86.90

Ln(age)* Current smoker

-0.94

0

0.00

N/A

N/A

N/A

Ln(age)* Family history of ASCVD

-1.53

0

0.00

N/A

N/A

N/A

Individual sum

N/A

N/A

142.04

N/A

N/A

119.22

Mean(Coefficient×Value)

N/A

N/A

140.68

N/A

N/A

117.26

Baseline survival

N/A

N/A

0.97

N/A

N/A

0.99

Estimated 10-year risk (%)

N/A

N/A

11.0

N/A

N/A

10.1

Abbreviations: ASCVD, atherosclerotic cardiovascular disease; SBP, systolic blood pressure; HDL-C, high-density lipoprotein cholesterol; Ln, natural logarithm; N/A, covariate was not included in the equation; –, this value was not included in the example.

SI conversion factors: To convert total cholesterol and HDL-C to mmol/L, multiply by 0.0259.

*Individual Example Value: The natural log of continuous covariates, and 0 or 1 for category covariate, interaction terms are the products of the natural log of age multiplied by the natural log of other continuous covariates, or by the value of category covariates.

Coefficient × Value: The product of the coefficient multiplied by the individual example value.

Supple table 2. Predicting an Individual’s 10-year Risk for a First ASCVD Event

We assume an individual aged 60 years with untreated SBP 130 mm Hg, total cholesterol 210 mg/dL, HDL-C 55 mg/dL, waist circumference 80 cm, nonsmoking, diabetes, living in urban area of northern China, and without family history of ASCVD.

For the equations, the values for age, SBP, total cholesterol, HDL-C, and waist circumference are log transformed. Current smoker, diabetes, and family history of ASCVD are dichotomous variables with 1 for “Yes” and 0 for “No”. Geographic region is dichotomous variable with 1 for “Northern China” and 0 for “Southern China”. Urbanization is also dichotomous variable with 1 for “Urban” and 0 or “Rural”. Interaction terms are the products of the natural log of age and the natural log of other continuous variables, or the value of categorical variables. See the “Individual Example Value” column in Supple Table 1.

These values above are multiplied by the coefficients from the gender-specific equations (“Coefficient” column in Supple Table 1), and the results are in the “Coefficient × Value” column in Supple Table 1. The sum of the “Coefficient × Value” column is then calculated for the individual in each gender group, and shown as “Individual sum”.

The estimated 10-year risk of the first ASCVD event is formally calculated as formula below:

(formula could be seen in the Attachment)

S10 is the survival rate for ASCVD at 10 years (“Baseline survival” in Supple Table 1), IndX’B is gender-specific “Individual sum” in Supple Table 1, MeanX’B is gender-specific overall mean “Coefficient × Value” sum, which is shown as “Mean (coefficient × value)” in Supple Table 1.

Below is the equation using men as an example to estimate the 10-year ASCVD risk:

which equals to a 11.0% probability of the first ASCVD event occurring within 10 years.

With regard to the risk levels classification you mentioned, indeed, in our study, we divided the subjects' risk of developing ASCVD for the first time in 10 years into three levels: low risk (<5%), medium risk (5-10%), and high risk (>10%) on the basis of subjects’ risk scores. We classify the risk levels according to the Chinese Guideline on the Primary Prevention of Cardiovascular Diseases [2], in order to achieve early intervention on high uric acid to reduce the 10-year risk of ASCVD.

For male population, 69.2% were low risk, 22.1% were medium risk, and 8.8% were high risk, and they were 94.3%, 5.1% and 0.6% among female, respectively. Male’s median 10-year ASCVD risk scores for each risk level were 1.79 (Low risk), 6.86 (Middle risk) and 12.68 (High risk), while female’s were 0.84 (Low risk), 6.35 (Middle risk) and 12.6 (High risk), respectively. Meanwhile, significant differences were found among risk levels group in both male and female population.

  • Yang X, Li J, Hu D, Chen J, Li Y, Huang J, Liu X, Liu F, Cao J, Shen C et al: Predicting the 10-Year Risks of Atherosclerotic Cardiovascular Disease in Chinese Population: The China-PAR Project (Prediction for ASCVD Risk in China). Circulation 2016, 134(19):1430-1440.
  • Chinese Society of Cardiology of Chinese Medical Association, Cardiovascular Disease Prevention and Rehabilitation Committee of Chinese Association of Rehabilitation Medicine, Cardiovascular Disease Committee of Chinese Association of Gerontology and Geriatrics, Thrombosis Prevention and Treatment Committee of Chinese Medical Doctor Association. Chinese guideline on the primary prevention of cardiovascular diseases. Chinese Journal of Cardiology. 2020; 48: 1000–1038.
  1. The data provided in Tables 1 and 2, looks like descriptive data. If so, the total numbers for each sectional row should be equal to the other sectional rows for each column, but it is not the case for all of them. For example, in Table 1, the total number related to drinks (yes and no) is not equal to the total number associated with marriage (married, single, divorce) for the Non-HUA column. Based on the information provided in the first-row corresponding to the sample size for this column, the total number should be n=4,921. Please clarify otherwise.

Response: Thanks for your comments. In this study, a standardized questionnaires were used to collect data. There were some subjects did not choose to answer some questions directly for privacy or other reasons, but chose the "unknown" option. We did not show the proportion of these people in Table 1 and Table 2 before. According to your comments, we added the option of unknown to the Table 1 and Table 2, so that the data in Table 1 and Table 2 can correspond one by one.

Thank you for your kind attention and looking forwards to your favorable reply.

Yours sincerely,

Feilong Chen

Tao Xu

Department of epidemiology and statistics

Institute of Basic Medical Sciences, Chinese Academy of Medical Sciences & School of Basic Medicine, Peking Union Medical College

5, Dong dan san tiao

Beijing 100005

China

Tel: 86 10 69156408
